# Appropriate Stubble Height Can Effectively Improve the Rice Quality of Ratoon Rice

**DOI:** 10.3390/foods13091392

**Published:** 2024-04-30

**Authors:** Wenju Yang, Xu Mo, Yiming Zhang, Zihao Liu, Qingwen Tang, Jia Xu, Sujun Pan, Yue Wang, Guanghui Chen, Yajun Hu

**Affiliations:** 1Department of Agronomy, College of Agronomy, Hunan Agricultural University, Changsha 410128, China; 2290884922@stu.hunau.edu.cn (W.Y.); zhangyiming202401@163.com (Y.Z.); 2839609271@stu.hunau.edu.cn (Z.L.); 20021227@stu.hunau.edu.cn (Q.T.); xujia2021@stu.hunau.edu.cn (J.X.); pansujun@hunau.edu.cn (S.P.); 2Yuelu Mountain Laboratory of Hunan Province, Hunan Agricultural University, Changsha 410128, China; mx525@stu.hunau.edu.cn; 3The Key Laboratory of Crop Germplasm Innovation and Resource Utilization of Hunan Province, Hunan Agricultural University, Changsha 410128, China; wangyue@hunau.edu.cn

**Keywords:** ratoon rice crop, rice quality, stubble height, amylose content, head rice rate

## Abstract

Ratoon rice, the cultivation of a second crop from the stubble after the main harvest, is recognized as an eco-friendly and resource-saving method for rice production. Here, a field experiment was carried out in the Yangtze River region to investigate the impact of varying stubble heights on the grain quality of ratoon rice, as well as to compare the grain quality between the main and ratoon season. This study, which focused on 12 commonly cultivated rice varieties, conducted a comprehensive analysis assessing milling characteristics, appearance, and cooking quality. The results show that ratoon rice crops exhibited a higher milled rice rate and head rice rate compared to the main rice crops. Conversely, chalky rice percentage, chalkiness degree, and amylose content were lower in ratoon rice crops. Principal component analysis grouped eight relevant quality indicators of rice quality which were concentrated into three categories, with amylose content identified as the key indicator of rice quality for distinguishing between different stubble heights. Random forest results reveal a robust and significant correlation between appearance quality index and amylose content. Subordinate function analysis indicated that a stubble height of 30 cm resulted in optimal rice quality, with Lingliangyou 211 exhibiting the highest quality and Xiangzao Xian 32 the lowest. Overall, our study suggests that ratoon rice crops generally outperform main rice crops in terms of quality, with the optimal measurement at a stubble height of 30 cm. This study holds substantial importance for selecting appropriate stubble heights for ratoon rice crops and enhancing overall rice quality.

## 1. Introduction

Rice, as a crucial cereal crop for humanity, represents a staple food for half of the global population [1]. China is the largest producer and consumer of rice in the world, playing a significant role in ensuring global food security. However, the ongoing processes of industrialization and urbanization in China have led to a diminishing availability of arable land resources [2]. Increasing the frequency of rice harvests under the limited land resources is the key strategy to address this challenge. In the scenario of insufficient warmth and sunlight resources in two seasons, but with a surplus in one season, ratoon rice represents a noteworthy approach to increase the yield per unit area. Ratoon rice involves the activation of dormant buds on rice stubs left post-main season harvest, leading to the growth of ratoon shoots, which subsequently mature and are harvested to yield an additional rice crop [3,4]. This cultivation approach offers several advantages, including improved efficiency in time and labor, higher yields, and substantial economic benefits [3,4,5]. Southern China is the main region for ratoon rice cultivation, covering over one million hectares and producing an average grain yield of 4500 kg per hectare [6]. Furthermore, the annual increase in the cultivation area of ratoon rice in China suggests a growing adoption of this model in the southern regions. With a significant improvement in living standards, there is an increasing demand for higher-quality rice. Given that ratoon rice from the additional cropping season meets current market expectations, this cultivation model has promising prospects.

Rice quality assessment generally includes four aspects: milling quality, appearance quality, cooking and taste quality, and nutritional quality [7]. A previous study has demonstrated significant improvements in the rice quality indicators such as head rice rate, chalkiness, and taste value during the ratoon season compared to the main season and late-season rice of the double-cropping system [6]. Yuan et al. [8] found that both the milling and appearance qualities of rice in the ratoon season markedly exceed those of the main season. In a study by Zhang et al. [9], the results revealed that the amylose content, as well as the hardness, cohesiveness, adhesiveness, and resilience after cooking, were higher for the rice harvested in the ratoon season, while the protein content was lower when comparing the rice quality characteristics between the main season and ratoon season rice. Additionally, sensory evaluations indicated an overall higher acceptance of rice from the ratoon season compared that from the main season.

Various agricultural management practices, including tillage, fertilizer input, and planting density, have a significant impact on crop yield [10,11]. Stubble height is a key factor to affect the grain yield of ratoon rice [12,13]. Studies have indicated that a high stubble cutting level during the main crop stage increased the regeneration rate and yield of the ratoon crop compared to the low stubble cutting level [14]. An excessively high stubble can impede light penetration, negatively affecting lower leaf photosynthesis and plant growth. Harrell et al. [15] also reported that reducing the cutting height of the main crop increased the number of auxiliary buds regenerated from the basal nodes. Applying nitrogen fertilizer appropriately in the first season also promotes the growth of regrowth buds, thus rapidly expanding the population of ratoon rice [16]. The height of ratoon rice stubble could affect the nutrient transport and source–sink relationship in rice plants, and ultimately impact both the yield and quality of ratoon rice [17,18]. Therefore, maintaining an optimal stubble height ensures a high regeneration rate and an adequate light distribution among plants, which benefits the growth of ratoon rice.

Previous research on the quality of ratoon rice has mainly focused on comparing the differences in rice quality between the main and ratoon season rice. However, there has been relatively less exploration of the variances in quality traits between these seasons under the different stubble heights in the ratoon season. Additionally, the quality variations among rice varieties also have an impact on the quality of ratoon rice during the ratoon season. In this study, we assessed the quality of rice from both the main and ratoon seasons under three stubble heights and utilizing 12 rice varieties commonly cultivated in the Yangtze River region. Based on the correlation analysis, random forest model, principal component analysis, cluster analysis, and subordinate function analysis, this study comprehensively conducted the evaluation of multiple traits of rice quality and aimed to identify suitable stubble heights for ratoon rice.

## 2. Materials and Methods

### 2.1. Test Sites and Materials

The field experiment was conducted in Mingyue Village, Changsha County, Changsha City, Hunan Province, China (longitude 112.37° E, latitude 28.55° N). The experimental materials consisted of 12 rice varieties commonly grown in the Yangtze River Basin. This comprises six varieties of conventional indica rice, namely Zhongzao 39, Zhongjiazao 17, Xiangzaoxian 45, Xiangzaoxian 32, Yuzhenxiang, and Longkezao 1, as well as six varieties of indica hybrid rice, including Luliangyou 996, Zhuliangyou 819, Lingliangyou 942, Taiyou 553, Wufengyou T025, and Lingliangyou 211. The seeds were provided by the Rice Research Institute of Hunan Agricultural University.

### 2.2. Experimental Design

Randomized block design was employed for our field study. The experiment involved three different stubble height treatments: 20 cm (H20), 30 cm (H30), and 40 cm (H40). Each treatment plot covered an area of 12 m^2^, with a plant spacing of 25 cm × 18 cm, and three replications were established for each treatment. In the main season, a compound fertilizer (with an N:P_2_O_5_:K_2_O ratio of 15:15:15) was applied as the basal fertilizer at a rate of 750 kg/ha, and urea (containing 46% N) was applied at a rate of 150 kg/ha after the tillering stage. In the ratoon season, N fertilizer was applied 7 days before harvesting the main season crop and 3 days after harvesting at a rate of 130 kg urea per hectare as a sprouting fertilizer, respectively. Rice grain quality was determined 90 days after harvesting at maturity for each treatment.

### 2.3. Determination Items and Methods

#### 2.3.1. Determination of Seed Milling Quality

Referencing the national standard “GB/T 17891-2017 Premium Paddy Rice” [19], we conducted the weighing 25 g of paddy and then processed it using a huller (JLGJ-45, Taizhou Luqiao Jingao Grain Equipment Factory, Taizhou, China) to remove the husk, yielding brown rice. The brown rice rate was calculated from this procedure. Subsequently, the brown rice was milled using an inspection rice milling machine (JNNJ3B, Hangzhou Daji Optoelectronics Instrument Co., Ltd., Hangzhou, China) to determine the milled rice rate. For further quality assessment, a Wanshen (SC-E) rice appearance quality scanner (Hangzhou Wanshen Detection Technology Co., Ltd., Hangzhou, China) was employed to identify whole milled rice and calculate the head rice rate.

#### 2.3.2. Determination of Appearance Quality

A total of 30 g of milled rice was spread on a Wanshen (SC-E) rice appearance quality scanner (Hangzhou Wanshen Detection Technology Co., Ltd., Hangzhou, China) for the measurement of the length–width ratio, chalky grain rate, and degree of chalkiness of the whole milled rice. The parameters set by this quality scanner for calculating chalkiness are chalkiness area >30 pixels and chalkiness surface area >10%.

#### 2.3.3. Determination of Amylose Content and Gel Consistency

The ISO 6647-2 method was employed to determine the amylose content in rice [20]. Standard rice samples (GSB 11-3875-2021) [20] for amylose content determination were purchased from the China National Rice Research Institute (Hangzhou, China). Measurement of the gel consistency was conducted according to the rice gel elongation method. Briefly, 100 mg of rice flour was added into a test tube, then 0.2 mL of 95% ethanol phenolphthalein solution and 2.0 mL of 0.2 mol L^−1^ KOH solution were introduced, while the test tube was gently shaken. Subsequently, the test tube was placed in a boiling water bath and the mouth of the tube was covered with a glass bead. After an 8 min water bath, the tube was cooled on a rack for 10 min, and then a 20 min ice–water bath was conducted. Finally, the test tube was laid flat, keeping it at around 25 °C. After 1 h, the length of the rice gel was measured.

#### 2.3.4. Data Processing and Analysis

The original data were organized using Microsoft Excel 2016 software. The paired t-test was applied to analyze the effects of stubble height on rice quality (*p* < 0.05) using the SPSS statistics software (Version 26). Correlation analysis was conducted to reveal the relationships between rice quality indicators. Random forest models were conducted utilizing the “randomForest” and “rfPermute” packages in the R software version 4.0.5 to identify the primary factors influencing amylose content and gel consistency. Principal component analysis (PCA) was employed to evaluate the relationship between stubble height and rice quality. Clustering of rice varieties was conducted using the K-means algorithm and Euclidean distance in OriginPro 2023 (64 bit) Beta3. The comprehensive assessment of rice quality in both main and ratoon rice crops was carried out using the subordinate function method using GraphPad Prism 9. The calculation method for the subordinate function is as follows [21]:Xu = (X − Xmin)/(Xmax − Xmin)(1)
Xu = 1 − (X − Xmin)/(Xmax − Xmin)(2)

In the equation, X represents the measured value of a resistance index for the test sample, while Xmax and Xmin denote the maximum and minimum values of this index across all samples, respectively. If the measured indices exhibited a positive correlation with the rice quality, Formula (1) was employed to compute the subordination function value; conversely, Formula (2) was utilized for negative correlations. Subsequently, the subordination function values of each index for each sample were aggregated, and their average value was determined.

## 3. Results

### 3.1. Comparison of Milling Quality between Main and Ratoon Seasons Rice at Different Stubble Heights

The average value for brown rice rates across varieties under different treatments ranged from 77% to 79%, for the milled rice from 71% to 73%, and for the head rice from 49% to 58% (Figure 1). Notably, in comparison to the milled rice rate of rice from the main season (MS), the treatments H30 and H40 during the ratoon season were higher by 2.7% and 3.1%, respectively. The head rice rates for H20, H30, and H40 were significantly higher than those of the MS, by 7.4%, 7.9%, and 9.0%, respectively. Compared to the MS, no significant effects of stubble treatments on brown rice rate were observed. However, a significant difference was found among the different stubble treatments (Figure 1).

### 3.2. Comparison of Appearance Quality between Main and Ratoon Seasons at Different Stubble Heights

The average value for the length–width ratio of rice grains varied between 2.2 and 2.3 across different treatments and varieties, for the chalky grain rate ranging from 28% to 43%, and for the degree of chalkiness varying between 8% and 13%. Considerable variability in the three appearance quality indicators was evident among the varieties, particularly with notable distinctions in chalky grain rate and chalkiness degree (Figure 2). A significant decrease was observed in the chalky grain rate and chalkiness degree of rice during the ratoon season compared to the main season. Additionally, no distinction was noted for the rice length–width ratio between main and ratoon season rice (Figure 2).

### 3.3. Comparison of Cooking Quality between Main and Ratoon Seasons Rice at Different Stubble Heights

The average value for amylose content in rice across different treatments and varieties ranged from 17% to 20%. Specifically, for H20, H30, and H40, there was a significant decrease of 2.7%, 3.0%, and 2.6%, respectively, compared to those of the MS. The average value of gel consistency of rice from different treatments and varieties varied from 52 to 64 cm. No significant difference in gel consistency was observed between the main and ratoon seasons rice. However, the gel consistency for both the H30 and H40 treatments was lower than that for the H20 treatment (Figure 3).

### 3.4. Relationships between Rice Quality Indicators

Several indicators exhibited significant correlations among milling, appearance, and cooking quality parameters, and only the head rice rate and gel consistency had limited correlations with other rice quality parameters (Figure 4). Amylose content significantly positively correlated with chalky grain percentage and degree of chalkiness, while it significantly negatively correlated with the length–width ratio of rice. Random forest modeling confirmed that the three appearance indicators are the most important influencing parameters on the amylose content, explaining 59% of its variance (Figure 5A,B). The chalky rice percentage was selected as the sole significant factor to affect the gel consistency, although its explanatory power was limited according to the random forest model (Figure 5C,D).

### 3.5. Principal Component Analysis of Rice Quality between Main and Ratoon Seasons Rice at Different Stubble Heights

Principal component analysis (PCA) was conducted to analyze the quality traits of rice from different treatments across varieties. The results show that the cumulative contribution rate of PC1 and PC2 was 90% (Figure 6). Specifically, PC1 and PC2 accounted for 60% and 30% of the total variance, respectively. The four stubble height treatments could be divided into three categories: H20 formed one category, located in the second quadrant; MS formed another category, located in the third quadrant; H30 and H40 formed a third category, located in the fourth quadrant. The variable loadings on PC1 indicated that brown rice rate, length–width ratio, and chalky grain rate significantly influence the assessment of rice quality. The variable loadings on PC2 suggested that the amylose content and milled rice rate exhibited high characteristic values, underscoring their pivotal roles in affecting the evaluation of rice quality.

### 3.6. Systematic Cluster Analysis of Rice Quality across Twelve Rice Varieties

By employing systematic cluster analysis, the twelve rice varieties were grouped into four categories (Figure 7). The initial category featured S12, distinguished by inferior milling quality yet commendable appearance and cooking quality. The second category encompassed S1, S4, S5, and S7, displaying superior milling quality but comparatively poorer appearance and cooking quality. The third category incorporated S3, S6, and S10, characterized by relatively subpar milling, appearance and cooking quality. Lastly, the fourth category comprised S2, S8, S9, and S11, exhibiting relatively inferior milling quality but superior appearance and cooking quality.

### 3.7. Subordinate Function Evaluation of Rice Quality between Main and Ratoon Seasons Rice at Different Stubble Heights

The subordinate function analysis method was employed to conduct a comprehensive evaluation of rice quality across various treatments and varieties. The results reveal that H30 exhibited the highest average subordinate function value (0.59), closely followed by H20 (0.58), while MS and H40 had the lower average values (0.56) (Figure 8A,B). This implies that optimal rice quality was observed when the stubble was left at a height of 30 cm. Among the varieties, S8 (Ling Liang You 211) recorded the highest subordinate function value (Figure 8C,D).

## 4. Discussion

The stubble height of ratoon rice can impact the nutrient absorption and accumulation of the rice plants. Maintaining an appropriate stubble height is essential for the healthy growth of rice plants, facilitating nutrient absorption and translocation [17]. Previous studies on the rice quality from ratoon seasons at different stubble heights have yielded varied results. For example, Yao et al.’s [22] study suggested that stubble height exerts a considerable impact on the milling quality of ratoon rice, with taller stubble heights proving advantageous for enhancing its milling quality. Xu et al. [18] examined the effects of high and low stubble heights on the quality of 18 approved three-line hybrid rice varieties. They observed that, compared to low-stubble treatments, high-stubble treatments significantly improved gel consistency in ratoon crops. In our study, by evaluating a range of rice quality parameters across twelve rice varieties at various stubble heights, we found that both milling and appearance quality were superior during the ratoon season compared to the main season. Additionally, we observed a decrease trend in amylose content and lower gel consistency in the ratoon season when compared to the main season.

Milling involves a combination of various unit operations to produce well-milled rice from raw rice. The findings of this experiment revealed that the brown rice rate in H30 and H40 treatments significantly surpassed that of MS, while no notable differences were observed in the milled rice rate across varying stubble heights. Furthermore, the head rice rate in H20, H30, and H40 treatments significantly exceeded that of MS, suggesting the enhanced processing quality of ratoon rice compared to main season rice, with the most pronounced improvement noted in the head rice rate. It is worth noting that the head rice rate stands out as the most crucial indicator of milling quality [23]. In a study conducted by Liu et al. [24], which compared the milling quality, appearance quality, and other aspects across twenty rice varieties between main and ratoon season rice, it was found that the milled rice rate and head rice rate of ratoon rice surpassed those of main season rice. This indicates an overall superior rice quality in the ratoon crop compared to the main season crop.

Appearance quality directly affects the market value of rice and is linked to the head rice yield. The findings of this study reveal that, during the ratoon season, all rice varieties exhibited lower rates of chalky grains and reduced chalkiness compared to the main season, with decreases ranging from 14% to 16% and from 3.5% to 4.6%, respectively. This result was similar to a previous study conducted in the southern U.S., where ratoon rice exhibited better appearance quality than the main crop [25]. Previous research has indicated that elevated nighttime temperatures can result in higher rates of chalky grains and chalkiness, along with a decline in milling quality [26]. Moreover, high temperatures during the grain-filling phase significantly impact rice quality, particularly in the formation of chalky grains [27,28]. The observed enhancement in the appearance quality of rice during the ratoon season may be attributed to several factors. Firstly, during the grain-filling stage of the ratoon season, both daytime and nighttime temperatures are lower compared to those during the grain-filling stage of the main season. At the study site, the average daytime temperature during the heading stage of the ratoon rice in September is 8 °C lower than that in July during the heading stage of the main season rice, with nighttime temperatures averaging 5 °C lower. Additionally, alterations in sugar composition within the carbohydrate metabolism pathway induced by ratoon rice cultivation may contribute to this improvement. Specifically, changes in starch composition, such as the amylose to amylopectin ratio, may lead to a reduction in both the quantity and surface area of dispersed starch granules within the grain endosperm [29,30]. These alterations ultimately influence the formation of chalky grains.

Amylose content serves as a critical parameter in assessing rice quality, exerting significant influence on cooking attributes and palatability [31]. Lower amylose content tends to result in stickier rice with improved texture [23]. Liu et al.’s [24] study also revealed that the amylose content in most ratoon rice varieties was lower than that in the main season. Moreover, research by Hu et al. [32] suggested that rice with greater gel consistency tends to be softer, whereas those with lesser gel consistency tend to be firmer. Additionally, Yuan et al. [8] investigated rice quality across 12 different locations and noted a decrease in gel consistency during the ratoon season compared to the main season. The results of our experiment demonstrated a reduction trend in amylose content and gel consistency by 2.6% to 3.0% and 4.8 to 11.9 cm, respectively, in samples H20, H30, and H40 compared to MS. Generally, indica rice with high amylose content is typically used for manufacturing rice noodles. However, for direct cooking, people in East Asia may prefer a more elastic, sticky texture and less hardness. Therefore, ratoon rice is favored for its cooking characteristics over main season rice in southern China.

Principal component analysis is a statistical technique used for dimensionality reduction, aimed at simplifying multiple variables into a few comprehensive variables based on their interrelations. Through PCA, the various treatments were categorized into three distinct groups: H20, MS, and H30/H40. Notably, amylose content exhibited significant eigenvalue, underscoring its pivotal role in assessing rice quality. Consequently, future research endeavors may consider selecting representative indicators derived from amylose content for preliminary evaluations of rice quality. Therefore, in subsequent related research, representative indicators can be selected from amylose content for a preliminary evaluation of rice quality. In our study, we also employed the random forest model to analyze the correlation between amylose content and milling as well as appearance quality indicators. The results demonstrated a strong correlation between appearance indicators and amylose content, suggesting that rice selection tailored to diverse cooking requirements can be achieved through the evaluation of rice appearance.

The method of subordinate function evaluation represents a flexible and effective approach for addressing various data decision-making challenges. Cluster analysis, on the other hand, offers a comprehensive, objective, and scientifically sound means of classifying test samples into distinct groups for evaluative analysis. This analytical method aims to group similar observations or data points together to form clusters. Through the application of subordinate function evaluation and cluster analysis to eight rice quality-related indicators across twelve rice varieties, it was determined that maintaining a stubble height of 30 cm for ratoon rice cultivation can significantly enhance rice quality. Notably, the cultivar Ling Liangyou 211 exhibited the highest quality among the tested varieties.

## 5. Conclusions

There exists a multitude of indicators for assessing rice quality, making it challenging to ascertain the superiority or inferiority of rice based solely on a single indicator. Therefore, integrating various indicators and employing diverse evaluation methods are necessary for a comprehensive assessment of rice quality. Random forest modeling, principal component analysis, subordinate function method, and cluster analysis are useful tools to explore the relationships among various indicators of rice quality. In this study, we compared and analyzed the variations in rice quality among twelve rice varieties during both the main and ratoon seasons at varying stubble heights. Our analysis encompassed the assessments of processing quality, appearance quality, and cooking quality between rice harvested during the main season and that during the ratoon season at different stubble heights. Overall, our findings suggest that maintaining a stubble height of 30 cm during the ratoon season can effectively enhance rice quality.

## Figures and Tables

**Figure 1 foods-13-01392-f001:**
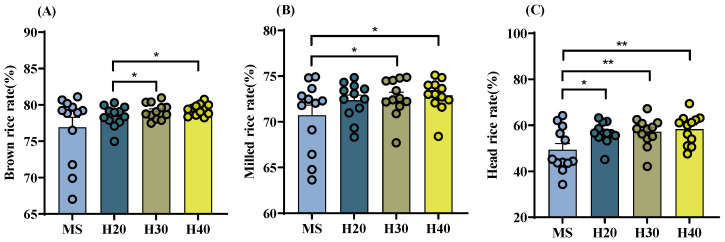
Milling quality of rice with different stubble heights in the main and ratoon rice crops. (**A**). Brown rice rate; (**B**). Milled rice rate; (**C**). Head rice rate; MS. Main Season; H20. Stubble height 20 cm; H30. Stubble height 30 cm; H40. Stubble height 40 cm; * *p* < 0.05; ** *p* < 0.01.

**Figure 2 foods-13-01392-f002:**
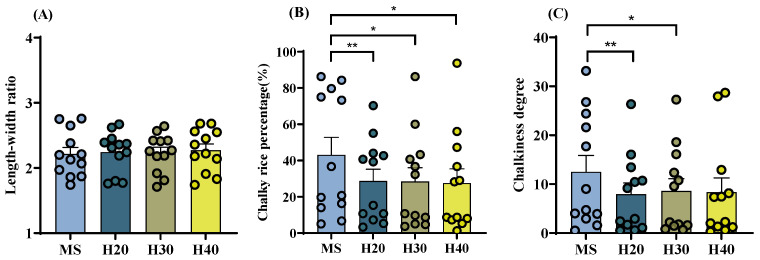
Appearance quality of rice with different stubble heights in the main and ratoon rice crops. (**A**). Length–width ratio; (**B**). Chalky grain rate; (**C**). Chalkiness degree; MS. Main Season; H20. Stubble height 20 cm; H30. Stubble height 30 cm; H40. Stubble height 40 cm; * *p* < 0.05; ** *p* < 0.01.

**Figure 3 foods-13-01392-f003:**
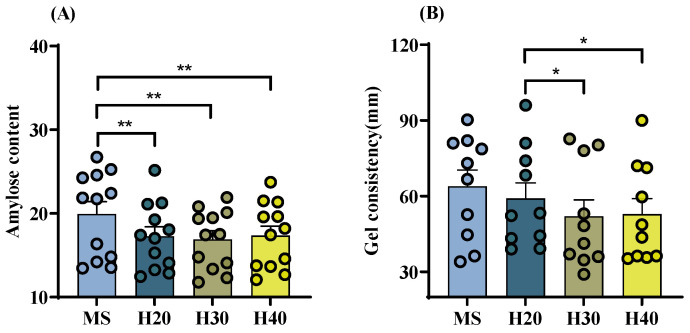
Cooking quality of rice with different stubble heights in the main and ratoon rice crops. (**A**). Amylose content; (**B**). Gel consistency; MS. Main Season; H20. Stubble height 20 cm; H30. Stubble height 30 cm; H40. Stubble height 40 cm; * *p* < 0.05; ** *p* < 0.01.

**Figure 4 foods-13-01392-f004:**
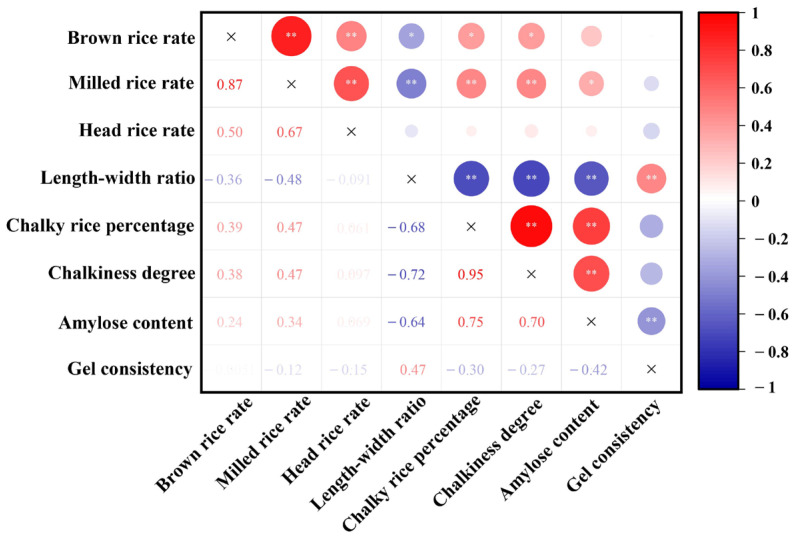
Correlation analysis of rice quality with different stubble heights in the main and ratoon rice crops. * *p* < 0.05, ** *p* < 0.01.

**Figure 5 foods-13-01392-f005:**
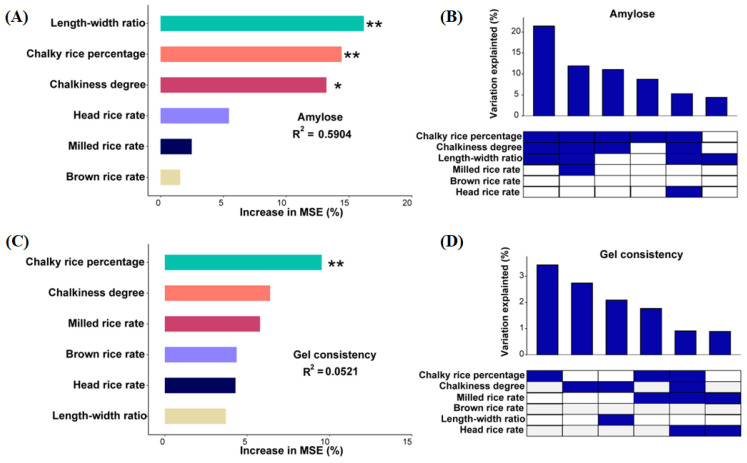
The relative importance (percentage of increase of mean square error, MSE) of milling and appearance quality variables on the amylose content (**A**) and gel consistency (**B**) based on full model of random forest analysis. Sparse model of random forest methods to predict the effect of milling and appearance quality variables on the amylose content (**C**) and gel consistency (**D**), respectively. * *p* < 0.05, ** *p* < 0.01.

**Figure 6 foods-13-01392-f006:**
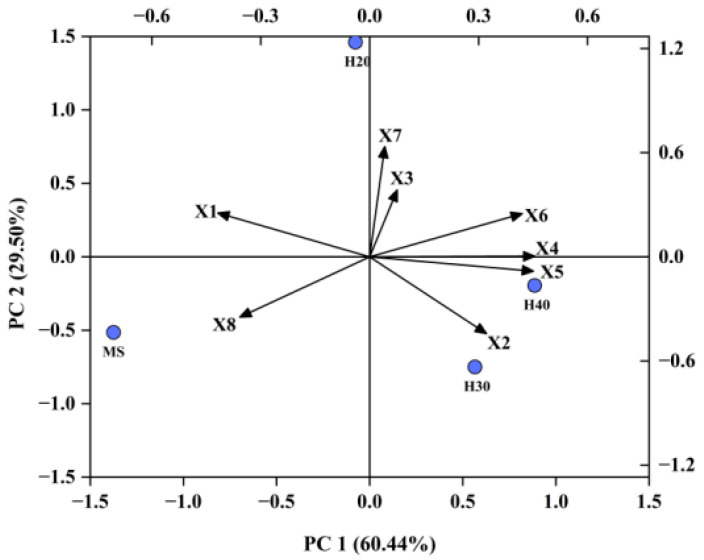
Principal component analysis of 8 indicators related to rice quality. X1. Brown rice rate; X2. Milled rice rate; X3. Head rice rate; X4. Length–width ratio; X5. Chalky grain rate; X6. Chalkiness degree; X7. Amylose content; X8. Gel consistency.

**Figure 7 foods-13-01392-f007:**
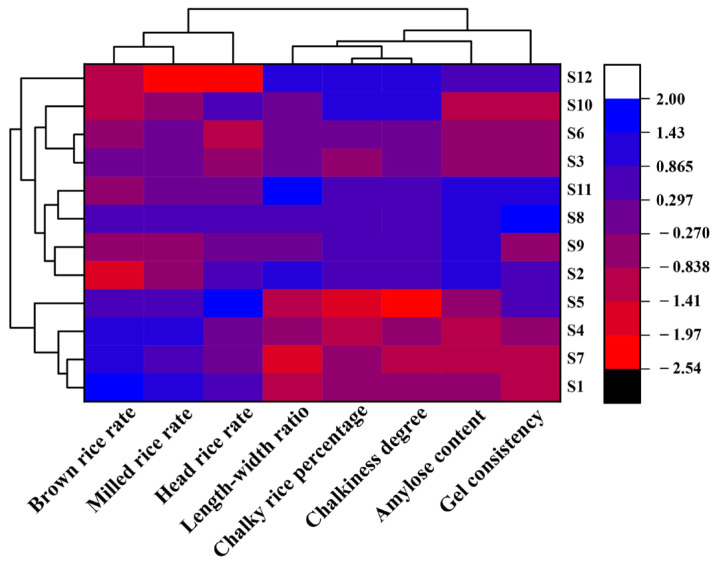
Cluster analysis of 12 rice varieties. S1. Zhongjiazao 17; S2. Longkezao 1; S3. Zhuliangyou 819; S4. Luliangyou 996; S5. Zhongzao 39; S6. Lingliangyou 942; S7. Xiangzaoxian 32; S8. Lingliangyou 211; S9. Xiangzaoxian 45; S10. Wufengyou T025; S11. Taiyou 553; S12. Yuzhenxiang.

**Figure 8 foods-13-01392-f008:**
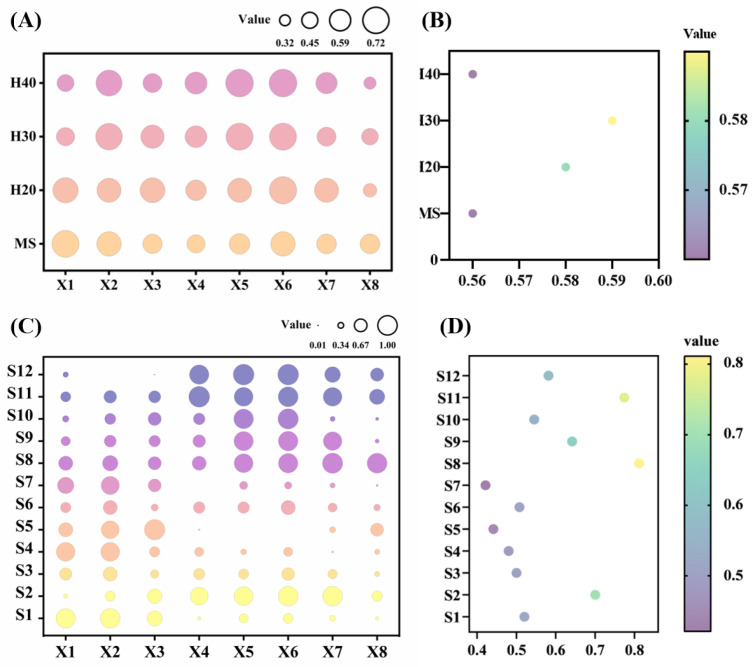
Subordinate function evaluation of rice quality of the main and ratoon rice crops with different stubble heights and varieties. (**A**,**B**). Evaluation of subordinate function of different stubble height between the main and ratoon rice crops; (**C**,**D**). Evaluation of subordinate function between varieties; X1. Brown rice rate; X2. Milled rice rate; X3. Head rice rate; X4. Length–width ratio; X5. Chalky rice percentage; X6. chalkiness degree; X7. amylose content; X8. gel consistency. S1. Zhongjiazao 17; S2. Longkezao 1; S3. Zhuliangyou 819; S4. Luliangyou 996; S5. Zhongzao 39; S6. Lingliangyou 942; S7. Xiangzaoxian 32; S8. Lingliangyou 211; S9. Xiangzaoxian 45; S10. Wufengyou T025; S11. Taiyou 553; S12. Yuzhenxiang.

## Data Availability

The original contributions presented in the study are included in the article, further inquiries can be directed to the corresponding authors.

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
