# Peer review of "Appropriate Stubble Height Can Effectively Improve the Rice Quality of Ratoon Rice"

_foods, 2024, doi:10.3390/foods13091392_

Round 1

Reviewer 1 Report

Comments and Suggestions for Authors

Dear Authors,

I found the manuscript interesting, but I suggest some changes that I hope will help improve the completeness of the work.

The adoption of the Ratoon rice cultivation technique contributes to increasing the productivity of rice, in response to the decrease in the availability of arable land. However, in the manuscript I did not find any reference relating to the increase in the inputs used, in particular nitrogen, which could heavily influence the economic and environmental sustainability of the second harvest obtained.

Also with the aim of making the work more international with respect to the literature cited, I suggest to read and consider the approach adopted by Deligios et al., 2017 (doi: 10.1016/j.eja.2016.10.016).

Can Ratoon rice stubble height increase rice cropping system sustainability?

Author Response

# Comments from Reviewer 1

General comments:

1、I found the manuscript interesting, but I suggest some changes that I hope will help improve the completeness of the work.

Response: Thanks for the positive and constructive feedback! We made use of the suggestions to further improve the manuscript quality.

Specific comments:

1、The adoption of the Ratoon rice cultivation technique contributes to increasing the productivity of rice, in response to the decrease in the availability of arable land. However, in the manuscript I did not find any reference relating to the increase in the inputs used, in particular nitrogen, which could heavily influence the economic and environmental sustainability of the second harvest obtained.

Response: Thanks for the valuable advice. Indeed, there is a close relationship between the yield of ratoon rice and the amount of nitrogen fertilizer input. We added the introduction of the nitrogen fertilizer would enrich the presentation of ratoon rice.

“Applying nitrogen fertilizer appropriately in the first season also promotes the growth of regrowth buds, thus rapidly expanding the population of ratoon rice.”

Reference: HUANG Jin-wen, Jia-yi W-U, Hong-fei CHEN, et al. Optimal management of nitrogen fertilizer in the main rice crop and its carrying-over effect on ratoon rice under mechanized cultivation in Southeast China[J]. Journal of Integrative Agriculture, 2022, 21(2): 351-364.

2、with the aim of making the work more international with respect to the literature cited, I suggest to read and consider the approach adopted by Deligios et al., 2017 (doi: 10.1016/j.eja.2016.10.016).

Response: In response to this comment, we added more information from this reference in the introduction as follow: Various agricultural management practices, including tillage, fertilizer input and planting density, have a significant impact on crop yield.

3、Can Ratoon rice stubble height increase rice cropping system sustainability?

Response: I may not fully understand the meaning of this question, but I'll still attempt to answer it. The sustainability of a ratoon rice system involves a range of factors, such as soil quality, the application of chemical pesticides and fertilizers, and the preservation of biodiversity. The aim of the ratoon rice system is to achieve a balance between economic, social, and ecological benefits. However, in this study, we only focus on how stubble height affects rice quality, which could be considered one aspect of economic benefits. We didn't conduct a thorough assessment of the sustainability of the ratoon rice system in relation to stubble height. This could be a great idea to explore in the next phase of our research.

Reviewer 2 Report

Comments and Suggestions for Authors

This paper reports interesting and useful results on how stubble height improves rice quality of ratoon rice and should take minor revision.

Their findings suggest that maintaining a stubble height of 30 cm during the ratoon season enhanced rice quality.

I have only two comments:

1. In Abstract:  “Here field experiment was carried out in Yangtze River region to investigate the impact of varying stubble heights on grain quality of the main and ratoon rice.”  This is confusing.  This seems to imply you were studying stubble heights on main crop yields also, but yields were already formed for the main crop before stubble height was determined.  Please reword to make clearer.

2. In results and discussion sections you repeated report percentage values to the nearest 0.01.  Round these off to the nearest whole numbers in each case.

Author Response

# Comments from Reviewer 2

General comments:

1、This paper reports interesting and useful results on how stubble height improves rice quality of ratoon rice and should take minor revision.

Response: Thanks for the positive and constructive feedback!

2、In Abstract: “Here field experiment was carried out in Yangtze River region to investigate the impact of varying stubble heights on grain quality of the main and ratoon rice.”  This is confusing.  This seems to imply you were studying stubble heights on main crop yields also, but yields were already formed for the main crop before stubble height was determined.  Please reword to make clearer.

Response: Thanks for the comments! True, we amended the text as follows: A field experiment was conducted in the Yangtze River region to examine the influence of different stubble heights on the grain quality of ratoon rice, as well as to compare the grain quality between the main and ratoon seasons.

3、In results and discussion sections you repeated report percentage values to the nearest 0.01.  Round these off to the nearest whole numbers in each case.

Response: Thank you for the reviewer's comments. We have retained two significant figures for all reported percentages in the results and discussion sections.

Round 2

Reviewer 1 Report

Comments and Suggestions for Authors

Dear Authors,

in my opinion the article can be published in its current form and with the improvements made.

Author Response

Thanks for the positive and constructive feedback again!